# A Review on the Catalytic Remediation of Dyes by Tailored Carbon Dots

**Sabrina A. Beker** [1,*] , **Ivan Cole** [2] **and Andrew S. Ball** [1]

1   School of Science, RMIT University, Bundoora, Melbourne, VIC 3083, Australia; andy.ball@rmit.edu.au
2   Advanced Manufacturing and Fabrication, School of Engineering, RMIT University,
    Melbourne, VIC 3000, Australia; ivan.cole@rmit.edu.au
*   Correspondence: sabrina.beker@rmit.edu.au

**Abstract:** Water polluted with dyes has become a serious global concern during the twenty-first century, especially for developing countries. Such types of environmental contaminant pose a severe threat to biodiversity, ecosystems, and human health globally; therefore, its treatment is an utmost requirement. Advanced technologies including the use of nanomaterials represent a promising water treatment technology with high efficiencies, low production costs, and green synthesis. Among the nanomaterials, carbon dots, as a new class of carbon-based nanoparticles, have attracted attention due to their unique features and advantages over other nanomaterials, which include high water solubility, easy fabrication and surface functionalisation, excellent electron-donating ability, and low toxicity. Such properties make carbon dots potential nanocatalysts for the Fenton-like degradation of environmental pollutants in water. Although recent studies show that carbon dots can successfully catalyse the degradation of dyes, there are still limited and controversial studies on the ecotoxicity and fate of these nanoparticles in the environment. In this review, the authors aim to summarise the recent research advances in water remediation by technologies using carbon dots, discuss important properties and factors for optimised catalytic remediation, and provide critical analysis of ecotoxicity issues and the environmental fate of these nanoparticles.

**Keywords:** water remediation; dyes; carbon dots; Fenton-like catalysis; ecotoxicity

## 1. Introduction

Water pollution has always been a critical challenge globally. Approximately 663 million people do not have access to improved water sources, and more than 2.2 million people die every year due to the lack of clean water, mostly in developing countries. Clean and accessible water for all is within the 17 Sustainable Development Goals established by the United Nations, where improving water quality, wastewater treatment, and the safe reuse of water worldwide are key targets to be achieved by 2030 [1]. Consequently, the international scientific community is urged to research and develop innovative technologies and sustainable cost-effective solutions to ensure the safe treatment of water and the safe disposal or reuse of waste.

Worldwide, industry discharges an estimated 300–400 Mt of waste into water bodies every year which, if inadequately treated before its release, may adversely affect the ecosystem and human health [1]. Among the discharged industrial pollutants, wastewater containing dyes from textile, leather, and food industries are important pollution sources [2–4]. When coloured wastewater is discharged into an aquatic system, it causes a list of problems: firstly, it reduces light penetration in aquatic bodies, disturbing photosynthesis which consequently impacts floral and faunal health by lowering the dissolved oxygen concentration in the water [2]. Secondly, many commercial dyes including cationic dyes, such as crystal violet, basic fuchsin, methyl green and methylene blue, are of high concern due to their toxicity, mutagenicity, carcinogenicity, and other adverse human health effects [5–7]. Further, dyes can persist as environmental pollutants through different

trophic levels leading to biomagnification [3]. Despite efforts to improve the performance of wastewater systems to eliminate these pollutants and meet increasingly stringent regulations, conventional wastewater technologies can be limited, expensive, and ineffective for such hazardous contaminants (Table 1). Hence, to address these challenges, efficient and cost-effective wastewater treatment technologies are required to remove these industrial pollutants from water before its safe release into the natural environment.

Over the past few decades, advanced oxidation processes (AOPs) have presented potential applications for dye containing wastewater treatments [8]. Advanced oxidation processes offer the advantage of potentially destroying organic pollutants instead of simply separating them from the water. The AOPs comprise a series of methods including ozonation, photocatalysis, electrochemical oxidation, Fenton, and Fenton-like processes [8,9]. These methods are based on physicochemical processes that can oxidise and potentially destroy organic contaminants due to the generation of highly non-selective reactive oxidising agents such as hydroxyl radicals ($\bullet$HO) [9]. Among the different AOPs employed, the Fenton reaction represents a promising high-performance water treatment based on the formation of $\bullet$OH by a catalytic reaction between $Fe^{2+}$ and $H_2O_2$, which can fully oxidise organic compounds [10,11]. Recently, Fenton-like reactions have emerged as promising alternatives to the traditional Fenton process. Fenton-like reactions mimic the Fenton reaction by replacing the addition of a $Fe^{2+}$ source with $Fe^{3+}$ source or other transition metals [12–14]. These reactions have demonstrated interesting and innovative possibilities for water treatment given their efficient performance under mild conditions, such as neutral pH, and for the possibility of using carbon-based catalysts [14,15].

**Table 1.** Existing treatment processes for the remediation of dye-containing water and their advantages and disadvantages.

| Name | Description | Advantages | Disadvantages | Ref. |
|---|---|---|---|---|
| **Physical treatment** | | | | |
| **Adsorption** | The method comprises adsorbents made of highly adsorptive materials, usually porous materials that can adsorb dye molecules. | Effective for a wide variety of dyes. Recyclable. Environmentally friendly. Easy operation. | Expensive. Difficult to separate the adsorbent from dye. | [16] |
| **Flocculation/Coagulation** | Various inducing agents are added to dye wastewater to destabilise the charged dye particles which will clump together. The clusters are removed following filtration. | Economical. Robust method. Easy to conduct. | Significant sludge generation which will need correct disposal. Not suitable for all types of dye effluents. High maintenance as it requires expensive chemicals and optimum pH conditions. | [17] |
| **Electrocoagulation** | The electrocoagulation of dyes process includes electrochemistry, coagulation, and flotation; thereby involving three distinct stages that incorporate chemical and physical phenomena. | Economical. Effective. Environmentally versatile. Low sludge production. | High operating and maintenance costs. High energy consumption. | [18] |
| **Nanofiltration/Ultrafiltration** | The filtration process can separate dye from clean water by filtering the dye wastewater through a nano (0.001 micron) or ultra-membrane (approximately from 0.002 to 0.1 microns) which retains dye particles. | High efficiency. Effective for any type of dye. | Costly. High energy consumption. High maintenance due to the constant membrane clogging problem. | [19,20] |
| **Reverse osmosis** | Reverse osmosis is a system where the dye wastewater is passed through an extremely thin membrane (0.0001 micron) under high pressure retaining the dye on one side and clean water on the other. | Very effective for decolourising and desalting dye wastewater. | Costly. Requires high pressure. | [21] |
| **Chemical treatment** | | | | |
| **Electrochemical oxidation** | There are two different pathways in which electrochemical oxidation operates: via direct oxidation where the contaminant is directly oxidised on the anodic surface; and via indirect oxidation where the transfer of electrons is mediated by oxidant species such as the hydroxyl radicals. | No sludge formation. Suitable for soluble and insoluble dye removal. | High electricity cost. Additional hazardous material production. | [22] |
| **Ion exchange** | The ion-exchange process comprises the removal of dyes via the strong affinities between the functional groups on ion exchange resins and charged dye molecules. The dyes are separated from wastewater due to the formation of strong linkages between the resins and the dye molecules. | Reusable method. | Selective to a limited number of dyes. | [16] |

**Table 1.** *Cont.*

| Name | Description | Advantages | Disadvantages | Ref. |
|---|---|---|---|---|
| **Fenton reaction** | This reaction involves the formation of hydroxyl radicals by a reaction between iron (II) and hydrogen peroxide ($H_2O_2$). The free radicals formed can fully oxidise dye molecules. | Effective for a wide range of dyes. | High iron sludge generation. Limited to acidic pH. Short lifetime of the free radicals. | [23] |
| **Biological treatment** | | | | |
| **Bioremediation** | Dye degradation by metabolic pathways or adsorption by living/dead biomass including bacteria, fungi, yeasts, algae, and plants. | Flexible method. Economical. Environmentally friendly process. | Not effective for non-biodegradable dyes. Lengthy process. Unstable system. | [3] |

The past decade has also seen a rapid advance in nanotechnology that has stimulated extensive research on the application of carbon-based nanomaterials in catalysis. Specifically, carbon dots (CDs) have attracted a great deal of attention due to their unique physicochemical properties that could find use in catalysis [24,25]. As a new category of carbon nanomaterials with very low toxicity, photoluminescence features, and rapid electron transfer properties [26], CDs show significant potential for application in AOPs as Fenton-like catalysts for environmental remediation [27]. When combined with metal oxides, CDs have proven to enhance their photocatalytic activities, showing they possess high surface charge and electronic conductivity [28–31]. Dyes, pharmaceuticals, and other complex molecules, such as methyl orange, p-nitrophenol, diclofenac, and gemfibrozil have been successfully degraded using this approach [28,29,32]; however, as with Fenton reactions, certain drawbacks pose limitations to their application, including dependence on external energy and generation of toxic sludges.

The interaction of CDs and $H_2O_2$ may be potentially useful for the catalytic remediation of dyes. Due to their size effect and active sites, the activation of $H_2O_2$ by CDs is likely to enhance the catalytic performance by generating more free radicals due to the rapid transference of electrons, resulting in an efficient catalytic system. This can be increased by tailoring the surface groups on CDs surface which may create a denser electronic cloud favouring electronic transfers. Nonetheless, this has not been fully explored from the remediation of dyes viewpoint.

Although many reviews to date have focused on other applications of CDs [33–36], to the best of our knowledge, there is no current review focusing on their use in catalytic oxidation for the remediation of dyes and their impact on the environment in terms of toxicity and microbiota effects. Therefore, this review is relevant and timely because it details important research being conducted into emerging carbon dots technologies and their impacts on eventual deployment and release in the environment.

With the rapid increase of CDs use and their potential discharge into the environment, it is of fundamental importance to critically assess the current literature for a better understanding of the potential opportunities and ecological risks when using such technology. In this review, we critically assess a broad range of studies on carbon dots as potential catalysts for environmental remediation while considering their ecotoxicological aspects, with a specific focus on current limitations and opportunities for catalytic improvement. The main aims of this review are (1) to discuss the potential of using carbon dots as catalysts in environmental remediation, (2) to identify the key properties of carbon dots, (3) to identity the factors influencing catalytic performance of carbon dots in aqueous media, (4) to discuss the ecotoxicological effects of carbon dots in model organisms, as well as impacts on aquatic bacterial communities, and (5) to discuss the current challenges and perspectives in this emergent field.

## 2. Carbon Dots

Carbon dots (CDs) have emerged as a central theme of many scientific studies due to their exceptional physicochemical properties and potential applications in several fields of science and technology, especially for environmental purposes. They were first discovered in 2004 as an impurity in the synthesis of single-walled carbon nanotubes using gel electrophoresis [37]. The name "carbon dots" was first introduced by Sun et al. [38], and represents a general term covering all carbon-based fluorescent nanoparticles. Conventionally, CDs are defined as a subclass of photoluminescent nanoparticles, with quasi-spherical morphology and the characteristic size of <10 nm comprising carbon, oxygen, and hydrogen atoms [39].

Typically, the surface of CDs is composed of polar groups containing carboxyl, hydrophilic hydroxyl, or amino groups. Their composition and surface chemical groups can result in crucial differences in several properties and therefore behaviour of these particles [40]. The chemical structures and main physicochemical properties of CDs are diverse, varying with the different synthesis approaches and type of precursors. The synthesis of

CDs can be classified into two approaches: top-down and bottom-up methods. Top-down synthesis routes involve breaking or cutting bulk carbon precursors into nanosized particles through laser ablation [41], electrochemical oxidation, and arc-discharge methods [42]. The drawback of these approaches, however, is that they require harsh reaction conditions, long processing times, and expensive materials and equipment [43]. Bottom-up approaches refer to carbonisation between small molecules and/or polymers precursors with low molecular weight that consist of functional groups such as hydroxyl, carboxyl, and amino groups. These approaches can be plasma treatment, microwave-assisted synthesis, templated routes, thermal decomposition, and solvothermal or hydrothermal methods [42]. The latter has become the most popular method due to its simplicity, low cost, low toxicity, and high efficiency. A major advantage of the bottom-up approach to CDs is the possibility to synthesise dots exhibiting specific properties, via the correct selection of a large variety of starting precursors.

The type of carbon dot precursors is another important factor to be considered during their synthesis. The sources can be divided into synthetic or natural products and their advantages and disadvantages must be considered.

### 2.1. Synthesis of Carbon Dots Using Environmentally Friendly Resources as Precursors

Synthetic molecules have been used traditionally as precursors for the bottom-up synthesis of CDs. Over recent years, citric acid (CA) has become one of the most popular small-molecule precursors of CDs and has been used by several authors for a wide range of applications [44–47], followed by glucose [48] and urea.

The mechanism of the formation of CDs from molecular precursors is an important, yet still unclear, phenomenon. Studies suggest that generally organic molecules usually undergo four stages in the formation of CDs: polymerisation, aromatisation, nucleation, and growth [49,50]. For example, when using monoethanolamine and citric acid as precursors, they react through intermolecular dehydration, which leads to the formation of large-sized polymer nanoparticles. When temperature is increased during the reaction, the polymer nanoparticles shrink due to continuous intramolecular dehydration. In this stage, a number of bonds such as C=C and C=N are formed, and aromatic clusters are concurrently produced inside the polymers (aromatisation) [50]. Nucleation stage of CDs takes place when the concentration of aromatic clusters of polymers reaches the critical supersaturation point. The formed nuclei develop by the enhanced degree of aromatisation of the polymers in the reaction [49].

Simultaneously, several functional groups (hydroxyl, carbonyl, carboxyl etc.) are added to the nanoparticle's surfaces. Finally, the ratio of polymers:dots decreases, leading to a smaller particle size of the CDs and less polycrystallinity [49,50]. A possible mechanism for the formation of CDs by this approach, proposed based on the literature reports of carbonisation of carbohydrates, is presented in Figure 1.

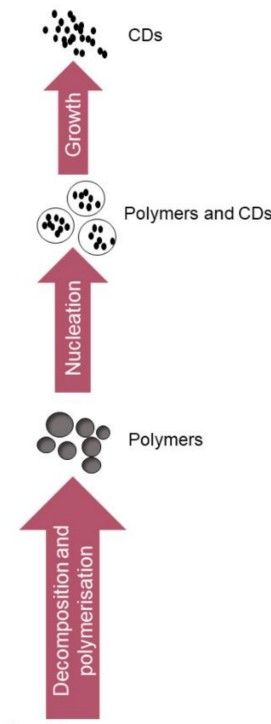

**Figure 1.** Possible mechanism for the formation of CDs using a bottom-up approach. Adapted from [50,51].

## 2.2. Sustainable Synthesis of CDs Using Natural Material as Precursors

Natural carbon sources are used for the preparation of CDs due to their low cost, environmentally friendly nature, and wide availability. A large variety of carbohydrates containing biomass, plant extracts, fruits, and kitchen wastes have been explored by researchers worldwide as a source of carbon which can be easily used for preparing CDs by subjecting them to thermal treatments [52] (Figure 2).

The precursors used vary widely and include honey [53], milk [54], peanut shells [55], egg shells [56], sugarcane bagasse [57], waste of wine fermentation [58], papaya [59], onion [60], lemon peel [61], *Manilkara zapota* fruits [62], turmeric [63], broccoli [64], and many more. Most of the preparation methods have followed the carbonisation of carbon containing precursors.

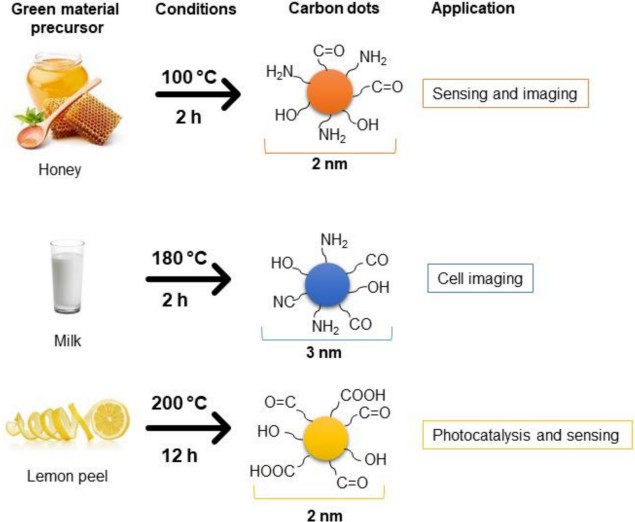

**Figure 2.** Schematic illustration of synthesis of carbon dots made of natural precursors. Adapted from [53,54,61].

A study conducted by De and Karak [65] reported a green and facile approach for the synthesis of water-soluble fluorescent CDs from banana juice. Green luminescent water soluble oxygenous CDs with an average size of 3 nm were synthesised by simply heating banana (*Musa acuminata*) juice at 150 °C for 4 h without using any surface-passivating and oxidising agent or inorganic salt. According to the authors, hydrolysis, dehydration, and decomposition of different carbohydrates takes place in the presence of ascorbic acid, resulting in soluble compounds such as furfural aldehydes, ketones, and several organic acids such as acetic, levulinic, and formic acids, etc. which act as acid resources for different catalytic reactions. Polymerisation and condensation of these products transforms them into different soluble polymeric products. Aromatisation and carbonisation then take place via condensation and cycloaddition reactions. Finally, the CDs are obtained by a probable nuclear burst of these aromatic clusters at a critical concentration at the supersaturation point [65].

### 2.3. Properties of Carbon Dots

Carbon dots show attractive and unique physical and chemical properties which enable them to be used for a broad range of applications. Most of the properties of CDs can be tailored depending on the precursor, synthesis method, and or functionalisation [66]. Such properties may be adjusted to increase as catalytic activity across many types of reactions, including photocatalysis, electrocatalysis, peroxidase-like catalysis, Fenton-like catalysis, and chemical catalysis [35,37,66,67]. In this section, the key physical and chemical properties of CDs are discussed in detail. Further, Figure 3 illustrates the generalised structure of CDs made of citric acid, urea, and iron (III).

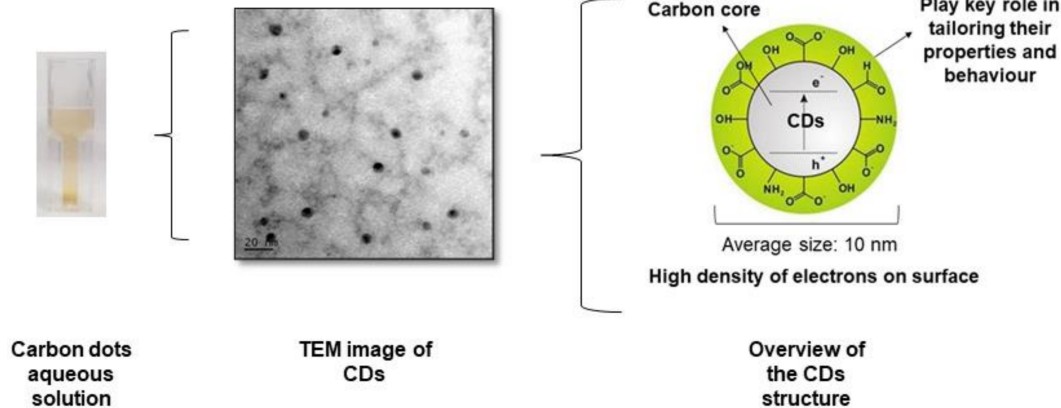

**Figure 3.** Carbon dots made of citric acid, urea, and iron (III) in aqueous solution, under the transmission electron microscope, and a schematic illustration of their structure.

### 2.3.1. Size

Generally, most CDs are less than 10 nm. The key factors affecting the size of the CDs have been studied extensively, including types of precursors, reaction time, solvent used, and temperature [68]. The reaction conditions are critical because they exert strong effects on the nucleation and growth processes. For example, reaction temperature and time can generally tune the size of CDs prepared by thermochemical decomposition. It has been reported that increased hydrothermal reaction temperatures decrease the average size of CDs from 20.35 nm (at 200 °C) to 6.48 nm (at 260 °C) [69]. A similar trend was observed when investigating the size evolution of CDs at different reaction temperatures; the hydrodynamic diameters of dots decreased from 177 to 79.7 to 42.1 to 5.3 nm as the temperature increased from 130 to 150 to 170 °C, respectively [49]. When varying reaction time from 2 h to 12 h and keeping the temperature at 200 °C, Papaioannou et al. [50] observed the formation stages of the CDs and consequently their decrease in size, from carbon particles of 400 nm at 2 h to nanoparticles of 2.8 nm at 12 h. Thus, the

CDs size can be controlled through the optimisation of reaction parameters which can offer uniform properties. Nanoparticle uniformity is important for many practical and functional applications. It is a critical parameter necessary to predict their behaviour in different environments.

Size is an important parameter influencing catalytic performance, as the concentration of surface-active sites can be dramatically boosted by shrinking particle sizes to the nanoscale [70]. Zhou et al. [38] observed that photocatalytic activity of the as-prepared CDs was found to increase as the size of the particles decreased. The photocatalytic degradation of organic dyes such as Rhodamine B (RhB) and methylene blue (MB) was significantly affected by the size of the CDs; neither RhB nor MB were degraded in the presence of 4.0–4.5 nm and 3.0–3.5 nm CDs; however, in the presence of smaller CDs (1.8–2.2 nm size), the dyes were completely degraded in 150 min [68].

Recently, CDs have emerged as a new type of carbon nanozymes showing excellent enzyme-like activity [35,71,72]. Along with other carbon-based nanomaterials, including carbon nanotubes and graphene oxide, CDs showed superior peroxidase-like activity to that of the natural enzyme horseradish peroxidase (HRP), confirming the promising potential of nanozymes to replace specific natural enzymes. For example, CDs showed higher peroxidase-like activity than graphene oxide (GO) and carbon nanotubes (CNTs) due to their size effect and enriched active sites [35].

### 2.3.2. Surface Functional Groups

Currently, surface functionalisation of CDs is an important and useful way to effectively modify their intrinsic surface state and consequently enhance their performance. In the absence of additional functionalisation, bare CDs generally contain abundant carboxylic groups on their surface [73]. Non-functionalised CDs usually lack diverse active sites due to the unicity of surface functional groups, which can limit their applications [73]. Consequently, functionalisation is a necessary strategy that endows CDs with unique properties, which can be performed by including non-metal and metal elements on their surface (Figure 4). In general, CDs can act as a reducing agent due to the reducing activity of functional groups, especially the hydroxyl group [37].

Among the non-metal dopants used for functionalisation, nitrogen, sulphur, and phosphorus are the most common elements [74], whilst the most used metals are copper, iron, zinc, and magnesium [75]. Doping heteroatoms such as nitrogen onto carbon materials may effectively tune their performance because of the ease of bonding with carbon atoms [76–78]; moreover, functionalisation with heteroatoms or metals will change the surface reactivity and adjust the band gaps of CDs, rendering the electron transfer within reactions more facile; thus, doped CDs possess a certain catalytic performance [79]. According to Li et al. [80], metal-ion doping reduces the surface function of CDs, adjusting their bandgap, facilitating electron transfer, endowing the metal-doped CDs with unique catalytic properties, increased binding selectivity, and high performance. A study conducted by Lin et al. [81] described that the five valence electrons from the N atoms in addition to the $Cu^+$ state increased the catalytic activity of N, Cu–CDs by changing the spin density and charge distribution of the carbon atoms in the dots, causing the energy gap to undergo drastic changes, enabling the adjustment of the CDs' physical and chemical properties.

In regards to metal-ion doping, metal ions have larger atomic radii than non-metallic heteroatoms and can accept and donate electrons more readily due to the nature of their electron orbitals (especially transition metal ions) [82]. With metallic ions doping into CDs, there may be more opportunity for changing charge density and charge transition forms between the metal ions and the carbon matrix via combining metal ions with the excellent electronic mobility of CDs. The electrons and unoccupied orbitals outside of metal ions facilitate them chelating with amino and carboxyl groups on the surface of CDs in the chemical reaction procedure, therefore extending the physicochemical properties of CDs and their catalytic performance [82,83]. Progress has been made on the application of CDs as solid supports for immobilisation of catalytically active nanoparticles or as enhancers of

photocatalytic and photoelectrochemical processes; however, to date, little attention has been paid to the direct use of CDs in catalysis.

Nitrogen-doped CDs (N–CDs) have attracted attention in catalytic applications because of their unique physicochemical properties; for example, a large number of defects at the surface may form more catalytic sites to create more effective catalytic activities [84,85]. This phenomenon has been observed for many photocatalytic and electrocatalytic applications [79,86–90]. For metal doping, it is believed that their introduction onto CDs will effectively influence the spin density and the charge distribution, which enhances the density of the catalytically active centres of CDs for binding redox species in catalytic reactions [82,84,91].

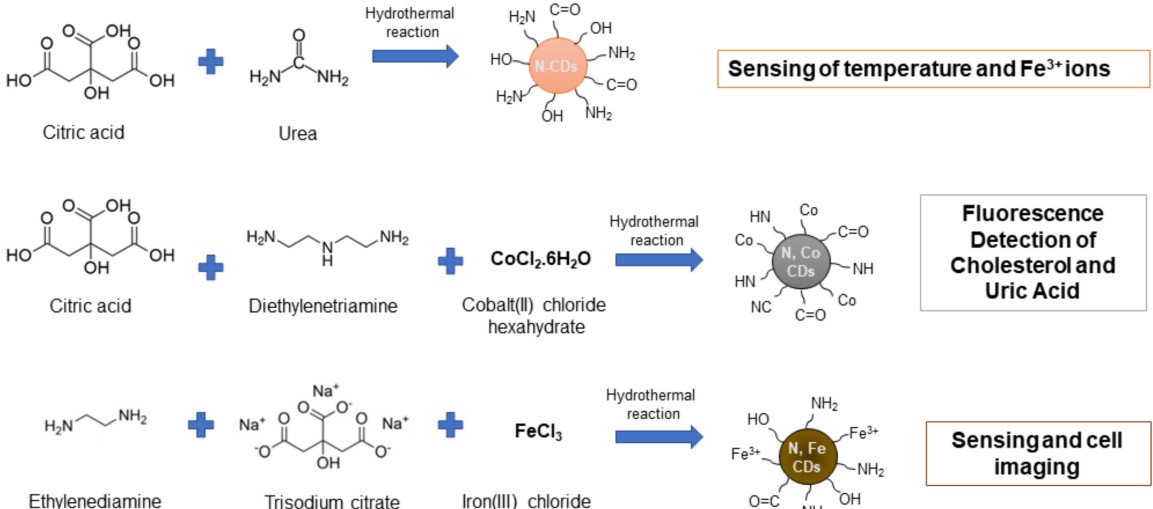

**Figure 4.** Schematic illustration of the preparation of various CDs made of carbon precursors and metal ions and their applications. Adapted from [92–94].

### 2.3.3. Charge Transfer Properties (Electron-Donating/Accepting Properties)

Significantly, CDs can act as electron donors and acceptors; this electronic property provides a powerful tool to fulfil the requirement of catalysis and other important applications. Due to superior electron transfer, along with the ability to trap and transfer electrons [95], CDs have been introduced into photocatalytic systems for further improving the photocatalytic activity of other materials, such as metal oxides and nanofibers [28,29,61]. Heterostructures of $TiO_2$ with CDs are found to exhibit improved charge separation along with transportation, ultimately leading to enhanced photocatalytic activity. Basically, upon UV irradiation, excitons (electron–hole pairs) are created over $TiO_2$ nanofibres and electrons are transferred to carbon quantum dots from $TiO_2$ conduction band, while holes remain in the valance band of $TiO_2$. This reduces the recombination of excitons and thus enhances the photocatalytic reaction [28]. This property and mechanism can be analysed since these electrons can lead to the generation of superoxide ion ($O_2^-$) upon reaction with oxygen at $TiO_2$–CQDs photocatalyst surface, and the holes produce hydroxyl radical (OH•) upon reaction with water [28].

Similarly, zinc oxide (ZnO) is a relevant semiconductor material of the II-VI group element and it has been used in heterojunction with CDs to improve photocatalytic activity [96]. The oxygenated functional groups on the CDs surface and their electron transfer properties represent an advantage of ZnO/CDs to separate photogenerated electrons and enhance photocatalytic activity for wastewater treatment [96] and dye molecules [97].

### 2.3.4. Water Solubility and Dispersibility

For CDs, water solubility and dispersibility can be a fundamental requirement for solution-processing and related applications as they can improve their physical and chemi-

cal interactions [39]. However, this property depends greatly on the CDs surface chemistry derived from precursors or generated during synthesis [39]. Most of the CDs are hydrophilic with excellent stability in solution which can be attributed to electrostatic stabilisation since their surfaces contain oxygenated functionalities. Many types of CDs have been reported as soluble in water to date [65,98,99]; nitrogen-doped CDs, for example, can show high solubility of more than $1 \, g \, mL^{-1}$ in solution, which presents an advantage in terms of bio-application [100]. Nonetheless, hydrophobic CDs have been prepared via pyrolysis of aliphatic amine and aliphatic aldehyde in order to be dissolved in organic solvents such as toluene, dichloromethane, and hexane [101]. In addition, hydrophilic CDs can be made hydrophobic by the covalent attachment of dodecylamine [39] and modifying them with other hydrophobic molecules [34].

Potential applications of CDs are greatly influenced by hydrophilicity and dispersibility. Some emerging areas make great use of such properties; for example, in biosensors, bioimaging, drug delivery, catalysis, and environmental applications [102]. However, given the hydrophilic properties of some CDs, they might have an unknown effect on natural environments, especially on aquatic environments and living organisms; therefore, this property should be considered when designing, using, and discarding CDs.

### 2.4. Characterisation Techniques Used to Analyse the Properties of Carbon Dots

The characterisation of CDs provides important information on their potential use for specific applications [103]. The catalytic behaviour of the CDs might be tailored according to their surface properties. Therefore, physicochemical characterisation of CDs is important and may contribute to understanding their role in applications such as the catalytic remediation of pollutants. Some basic surface information can be obtained from techniques commonly used in the laboratory (Figure 5). For example, key catalytic properties, such as size, morphology, and crystallinity of the CDs, can be characterised using transmission electron microscopy (TEM) [104]. The small size of CDs may provide a large specific surface area for a more convenient catalytic reaction [105]. For example, good electrocatalytic activity was observed on CDs to NiFe–LDH (ultrathin nickel−iron layered double hydroxide nanoplates), which was mainly attributed to the NiFe–LDH phase and further enhanced by the strong association of the LDH with CDs, which are small and exhibit excellent rapid electron transfer and electron reservoir properties. The dimensions of the CDs can be assessed by atomic force microscopy (AFM), dynamic light scattering (DLS), or by TEM.

Surface charge is a crucial property that is involved in catalysis, and CDs can be prepared with either a positive or negative surface charge. The surface charge can be measured using zeta potential ($\zeta$-potential or ZP). The ZP is basically the difference in potential between the dispersion medium and a stationary layer of fluid attached to the dispersed particles, and it is related to the electrical charge on the surface of nanoparticles dispersed in liquid media that depends on the particle itself and the surrounding medium [106]. It is widely known that the surface charge state of the catalyst is one of the most important factors for tuning catalytic performance in addition to surface active sites [107]. For example, catalytic reaction activity and selectivity can be tailored by the surface charge, resulting in an optimised catalytic activity.

The chemical surface can provide catalytic functionalities to the CDs; therefore, it is necessary to identify and quantify them. Based on the interaction of the infrared (IR) radiation with matter, Fourier transform infrared spectroscopy (FTIR) may be used for the identification and characterisation of the chemical structures of the CDs. The carboxyl, amino, and hydroxyl groups can be identified by FTIR and supported by X-ray photoelectron spectroscopy (XPS) analysis or by nuclear magnetic resonance (NMR) [108,109].

Significantly, CDs are widely known due to their peculiar and fascinating photoluminescent (PL) properties. Carbon dots present a strong blue emission which declines in the red region, typically characterised by dependence from excitation wavelength [39]. Generally, PL properties depend on the intrinsic nature and surface characteristics of the

CDs, such as surface chemistry, quantum confinement effects, and other factors; however, the PL mechanism is still unclear [34].

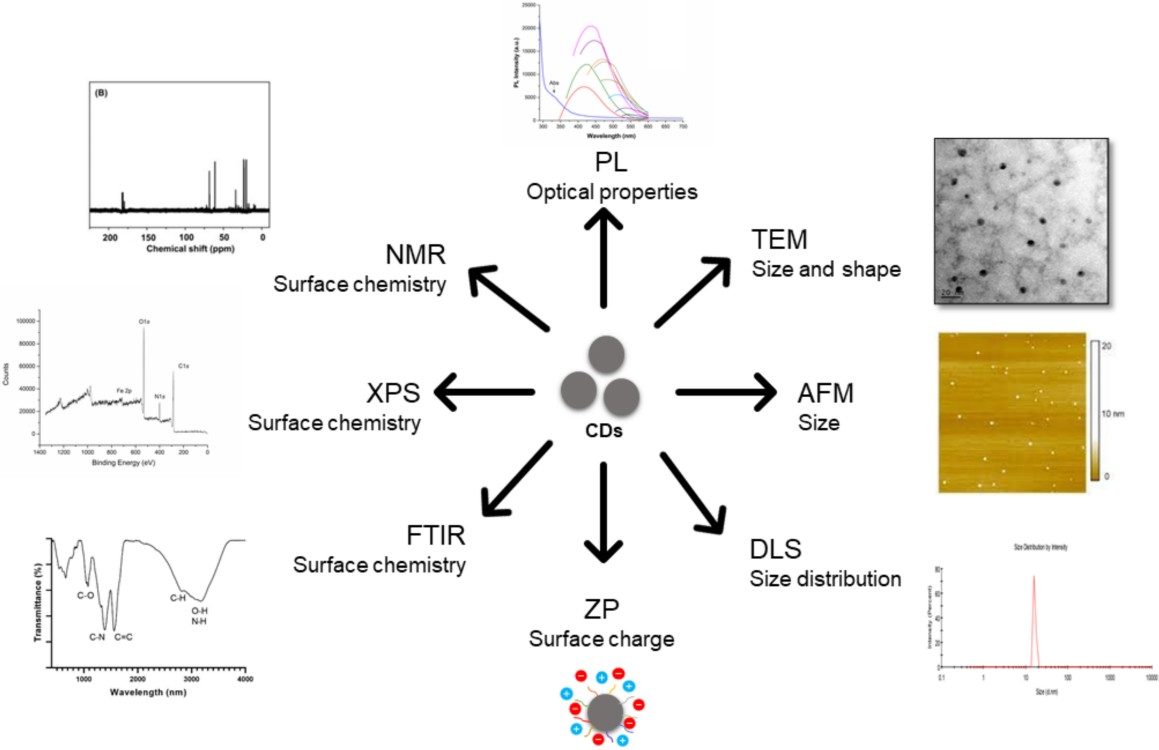

**Figure 5.** Physicochemical techniques employed for the characterisation of carbon dots.

Altogether, these pieces of information allow researchers to elucidate the range of catalytic use of the considered CDs and finally to evaluate the catalytic loading in terms of the referred functional groups, surface charge, size, etc.

*2.5. Mechanism Involved in the Catalytic Remediation of Dyes by Carbon Dots in the Presence of $H_2O_2$*

The Fenton-like mechanism is based on the production of hydroxyl radicals (•OH) and other free radicals to partly or completely decompose organic compounds without selectivity [10]. However, one of the main challenges faced by these reactions during catalysis is the short lifespan of free radicals, which is considered one of the most important factors in the feasibility of Fenton's reaction. For example, the •OH radical has shown the longest lifetime of ~2.7 μs, followed by 1.3 μs of superoxide anions ($O_2^{•-}$), and 1.4 μs of hydrogen peroxide ($H_2O_2$), in ambient air of water surface [110]. Meanwhile, singlet oxygen ($^1O_2$) was reported to have a lifetime of 2.80 s and a diffusion distance of 0.998 cm at 23 °C under 1.0 atm [111]. Most of the reactive oxygen species (ROS) cannot diffuse far from the site of their production due to their instability and reactivity. Therefore, the distance that free radicals diffuse is crucial for catalytic performance.

A stronger electrostatic affinity between the catalyst and the pollutant is likely to contribute positively to the catalytic reaction in the presence of $H_2O_2$. We speculate that due to the proximity of the dot and the organic compound caused by opposite charges of the compounds, the distance that the ROS generated by the decomposition of $H_2O_2$ over the surface of the catalysts must travel is dramatically reduced, thereby enhancing the catalytic performance of the system. Another important factor that might influence catalytic activity is the surface area of the catalyst, which is associated with its size. This is evident when materials made up of nanoparticles have a relatively larger surface area when compared to the same volume of material made up of bigger particles. A large surface along with the strong electrostatic attraction can determine the extent of the adsorption

and catalytic reaction. Consequently, catalysis can be improved by choosing nanosized particles, such as carbon dots, as catalysts since they show size of <10 nm, tuneable surface charge, and good water solubility/dispersibility.

Recently, it has been demonstrated that the surface charge of CDs contributed to the catalytic performance in the oxidation of 3,5,3′,5′-tetramethylbenzidine (TMB) [100]. The authors studied the peroxidase-like behaviour of two types of CDs: CACDs (citric acid-modified carbon dots) and PEI (poly(ethyleneimine)-modified carbon dots) with zeta potential measurements of $-10.8$ mV and 12.7 mV, respectively. Their results indicated that the peroxidase-like activity of CDs varied with the surface modification. Cationic amino-modified CDs (PEI–CDs) showed weak catalytic activity toward TMB due to their low affinity toward the positively charged substrate. In contrast, citrate-capped CDs (CA–CDs), which were negatively charged, tended to attract amino groups of TMB electrostatically and showed strong catalytic activity toward TMB.

In this context, functionalisation of CDs has recently been successfully applied in Fenton-like reactions for the catalytic remediation of dyes [112–115]. Beker et al. [113] proposed that free radicals are formed on the surface of functionalised CDs in the presence of $H_2O_2$ (Figure 6). The reaction could lead to the oxidation of pollutants such as dyes by the efficient production of hydroxyls, superoxide anions and other reactive oxygen species (ROS) formed during the Fenton-like catalysis.

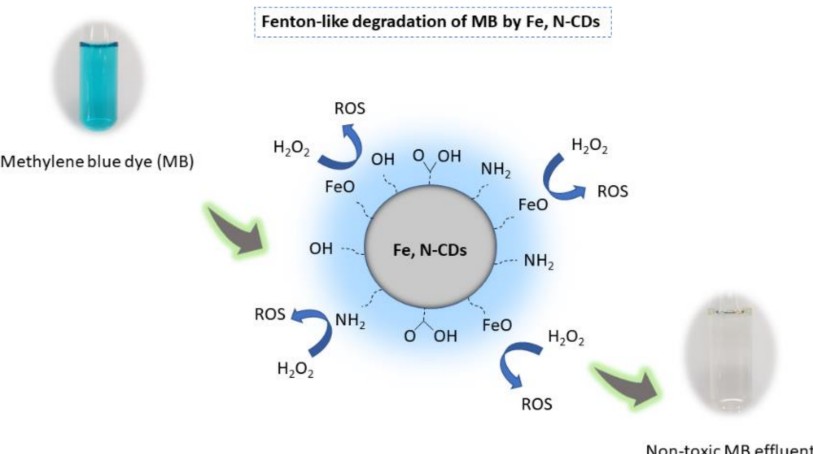

**Figure 6.** Illustration of the potential mechanism of CDs as Fenton-like catalysts [113].

*2.6. Factors Influencing the Performance of Carbon Dots on the Activation of $H_2O_2$*

The efficiency with which CDs activate $H_2O_2$ is determined by environmental factors such as temperature and pH, as well as $H_2O_2$ and catalyst concentrations. To reach the maximal degradation rate of pollutants, such activation reaction requires a specific set of conditions. As industrial effluents have a range of physicochemical properties, it is important to investigate the effects of significant parameters that may help to choose the most appropriate factors that influence the treatments efficiency. The main factors are listed and discussed below.

2.6.1. Effect of pH

Most AOPs are pH-dependent, being effective only at acidic pH, especially for Fenton processes. However, international pH standards of dye effluent discharge in the environment stand between pH 6 and 9 [2]. Therefore, catalysts that operate and accelerate in situ production of ROS at near neutral and slightly basic values are highly desired. Carbon dots have demonstrated the ability to perform and maintain their properties under different pH conditions in aqueous media [114]. For example, citric acid- and thiourea-derived CDs were tested as catalysts in the Fenton-like degradation of thymol blue dye at pH 6 to 9, and they exhibited excellent activity in a near-neutral solution [112].

Superoxide anions ($O_2^{\bullet-}$) were formed as key intermediate products in the reaction occurring in the CDs–NaOH system in alkaline solutions, while in near-neutral or acidic solution, such products were not detected [116]. The phenomenon could be attributed to the increased stability of $O_2^{\bullet-}$ (rate constant k =2 $M^{-1}$ $s^{-1}$), compared to the acidic or neutral solution (decay rate constant k =8.5 $\times$ 107 $M^{-1}$ $s^{-1}$). Similarly, the catalytic effect of cetyltrimethylammonium bromide-passivated carbon nanodots (CTAB–CDs) was found to decompose $H_2O_2$ effectively into $\bullet$OH and $O_2^{\bullet-}$ in a strongly alkaline medium, where they participated in the radical recombination reaction to form $^1O_2$ on the surface of CTAB–CDs [24]. This study suggests that the ROS formed during the catalytic reaction could be applied in the degradation of dyes.

A ternary combination of graphitic carbon nitride (g–$C_3N_4$), carbon nanodots (CDs), and ferrous ion (Fe[II]) was proposed to degrade MB in the presence of $H_2O_2$ [117]. Remarkably, no pH effect over the range of 3–9 was observed in terms of the degradation rate of MB. Although optimal degradation was achieved under acidic conditions (pH 3), with 66.8% MB removed within 1 h, the degradation rate for pH values from 5 to 9 ranged from 57.9 to 60.4%, with no significant differences observed, demonstrating that the CN/CDs/Fe(II) composite could be used in a wide pH range [117].

Iron and nitrogen co-doped carbon dots (Fe–N–CDs) were assessed regarding their enzyme-like activities, since some carbon nanomaterials can enhance the oxidation capacity of $H_2O_2$ [118]. They were shown to possess intrinsic peroxidase activity through catalysing the oxidation of 3,3,5,5-tetramethylbenzidine (TMB) to a deeper colour in the presence of $H_2O_2$. When compared with natural horseradish peroxidase (HRP), the Fe–N–CDs showed stable catalysis performance over a wider pH range, demonstrating comparable catalytic activity between pH 1 and 12 [118]. It seems that CDs can perform over a wide pH range, which is a significant advantage over other traditional catalysts that can operate under acidic conditions only. It is not clear in the literature how CDs keep their activity across a range of pH values; however, we speculate that the protonation or the deprotonation of functional groups over CDs does not directly affect their performance under catalytic activity. However, this might depend on their structure and surface charge state.

### 2.6.2. Effect of Temperature

The efficacy of the treatment of effluents by AOPs is known to be highly affected by temperature. Although the temperature used for Fenton-like processes is generally around 25–30 °C [10], the effect of higher temperatures has been investigated by many researchers due to the fact that (i) textile effluents are usually approximately 40–50 °C [2], and (ii) higher temperature could be beneficial to reduce the catalyst dose. For the performance of CDs as Fenton-like catalysts, temperature plays an important role. Our research group have evaluated the effect of different temperatures on the catalytic activity of prickly-pear-derived CDs towards the decolourisation of a mix of cationic dyes. Decolourisation was strongly influenced by temperature, with optimum decolourisation observed at the highest temperature (80 °C) [114].

However, studies have shown that a high temperature is not necessary for improved $H_2O_2$ decomposition. For example, the catalytic activity of CDs was tested for the oxidation of TMB at different temperatures (from 25 to 45 °C). The optimum temperature was 30 °C [119]. Similarly, at room temperature, CDs also showed high efficiency as catalysts for hydrogen peroxide decomposition, as showed by Bourlinos et al. [83] and Chen et al. [98]. The CDs appear to work under different temperatures, which is a highly desirable feature for environmental remediation.

### 2.6.3. Effect of Carbon Dots Concentration

The concentration of the catalysts also affects the catalytic reaction. Mohammadpour et al. [119] investigated the optimised conditions for the catalytic activity of CDs in the oxidation of TMB and found that the most appropriate concentration in the range 0.02 to 0.1 mg $mL^{-1}$ was 0.08 mg $mL^{-1}$. Similarly, the concentration of CDs was optimised for the Fenton-like

decolourisation of dyes, which ranged from 0.062 mg mL$^{-1}$ to 0.5 mg mL$^{-1}$, with the best result reported at 0.25 mg mL$^{-1}$ [114]. Other studies have reported similar catalytic behaviour of CDs in systems where it appears that the catalytic efficiency is controlled by stoichiometry [72]. It also seems that the CD concentration should be optimised for each individual reaction as there is no general concentration ratio to be applied.

*2.7. Comparison with Current Treatments Using Fenton-like Approach*

Various Fenton-like processes have been applied to treat dye-containing water. A comparison of recent studies that have used nanomaterials in general as Fenton-like catalysts in the degradation of dyes is presented in Table 2. Commonly, the treatments efficiency is estimated based on the reduction of the colour of the dye solutions which is calculated as follows: (%) = [(Ai − At)/Ai] × 100, where Ai is the initial absorbance of dye solutions and At is the absorbance of the dye at any time interval. Many metallic nanoparticles have been used to remediate dyes in the presence of $H_2O_2$; Su et al. [15] investigated the use of CuO nanowire arrays on copper mesh under neutral pH and ambient temperature for the Fenton-like degradation of Rhodamine B, crystal violet, methylene blue, and rhodamine 6G. The system efficiently degraded 100% of dyes in the presence of 10% $H_2O_2$ after 16 min, 20 min, 20 min, and 90 min, respectively. Similarly, Liu et al. [14] and Wolski et al. [120] used $Fe_3O_4$/$SiO_2$/C nanospheres and Au/ZnO nanoparticles, respectively, in addition to 5% of $H_2O_2$ to degrade approximately 90% of methylene blue dye. Although these systems showed efficient removal of such dyes, secondary pollutants would be generated by using metal-based nanoparticles in the presence of high $H_2O_2$ levels. Some Fenton-like catalysts may work efficiently only under UV light irradiation, which is also an additional cost that should be considered. Furthermore, harsh reaction conditions and the use of expensive precursors were employed for the synthesis of some catalysts [120,121]; this is rather disadvantageous from both economic and environmental points of view. Green synthesis concepts should be considered when developing and implementing catalysts to overcome such drawbacks.

Table 2. Use of different nanocatalysts and parameters in the decolourisation/remediation of dyes via $H_2O_2$ activation.

| Dye | Catalyst Dose | Experimental Conditions | | | | | Ref. |
|---|---|---|---|---|---|---|---|
| | | $H_2O_2$ Dose | pH | Temperature (°C) | Reaction Time (min) | Treatment Efficiency (%) | |
| **Rhodamine B, crystal violet, methylene blue, and rhodamine 6G at $2.5 \times 10^{-5}$ M** | CuO nanowire arrays on copper mesh | 10% | Neutral | Room temperature | 16, 20, 20 and 90, respectively | 100 | [15] |
| **Methylene blue at 100 mg $L^{-1}$** | $Fe_3O_4/SiO_2/C$ nanospheres at 100 mg $L^{-1}$ | 5% | 6.0 | 40 | 60 | 90 | [14] |
| **Methylene blue at 15 mg $L^{-1}$** | Au/ZnO nanoparticles at 0.36 mg $mL^{-1}$ | 5% | 7.5 | Room temperature | 120 | 90 | [120] |
| **Methyl red and methyl orange 10 mg $L^{-1}$** | CDs | 160 mM | 10 | 25 | 200 | 83 | [122] |
| **Methylene blue at 10 mg $L^{-1}$** | Photo-Fenton-like Barium M-hexaferrites nanoparticles SMN-NP (0.75 mg $mL^{-1}$) | 8 mM | 3 | Room temperature | 140 | 98.9 | [123] |
| **Methylene blue at 20 mg $L^{-1}$** | Fenton-like MnMgFe-layered double hydroxide (1.0 mg $mL^{-1}$) | ~1 M | 7 | 25 | 300 | 68 | [124] |
| **Methylene blue at 50 mg $L^{-1}$** | Photo-Fenton-like $CuFe_2O_4$ nanoparticles (0.2 mg $mL^{-1}$) | 20 mM | 5 | 25 | 80 | 52 | [125] |
| **Methylene blue at 20 mg $L^{-1}$** | Fenton-like $Cu_9S_5$ nanospheres (0.2 mg $mL^{-1}$) | ~60 mM | From 5.0 to 9.0 | 50 | 21 | 98 | [121] |
| **Methylene blue at 100 mg $L^{-1}$** | Fenton-like Iron oxide nanoparticles (M-NPs) (2.0 mg $mL^{-1}$) | 560 mM | 3.5 | 90 | 90 | 100 | [126] |
| **Methylene blue at 15 mg $L^{-1}$** | Fenton-like Gold nanoparticles deposited on ZnO (Au/ZnO) (1.05 mg $mL^{-1}$) | ~500 mM | 7.5 | Room temperature | 120 | 90 | [120] |

**Table 2.** *Cont.*

| Dye | Catalyst Dose | Experimental Conditions | | | | | |
|---|---|---|---|---|---|---|---|
| | | H$_2$O$_2$ Dose | pH | Temperature (°C) | Reaction Time (min) | Treatment Efficiency (%) | Ref. |
| **Methylene blue at 20 mg L$^{-1}$** | Fenton-like Fe, N–CDs (0.5 mg mL$^{-1}$) | 147 mM | 8 | 50 | 60 | 97.5 | [113] |
| **Mix of crystal violet, methyl green, basic fuchsin dyes at 15 mg L$^{-1}$** | *Opuntia* derived carbon nanodots (0.25 mg mL$^{-1}$) | 0.5% | 7.5 | 20 | 60 | 90 | [114] |

## 2.8. Environmental Impacts of Carbon Dots

### 2.8.1. Ecotoxicological Impacts of Carbon Dots

The toxicity of carbon-based nanomaterials may depend on several aspects, such as precursors used, concentrations, particle size, surface chemistry, durability, and aggregation; additionally, various routes of exposure are considered with different effects on organisms [127,128]. Although elemental carbon is of very low toxicity, carbon-based nanomaterials, such as carbon nanotubes and graphene oxide, can be harmful to microbes and animals by inhibiting microbial growth, reducing microbial diversity, destroying cell viability, and inducing apoptosis [127,129]. In terms of CDs, the literature has shown beneficial and detrimental effects over microalgae and bacteria [127,130,131], while CDs have increased *Chlorella vulgaris* growth by 21% when compared to the control group [131]; in contrast, the authors demonstrated toxic effects on the same microalgae by causing disruption in their photosynthetic system [130]. In one study, the toxicity of biochar-derived CDs was assessed using Microtox™ [132]. Three different feedstocks were used as precursors for the synthesis of biochar: *Picea, Molinia caerulea,* and *Elaeis guineensis* were used as precursors for the synthesis of CDs. According to aquatic toxicity analysis, the CDs were classified as highly toxic (*Picea* C-dots), moderately toxic (*Molinia caerulea* C-dots), and non-toxic (*Elaeis guineensis* C-dots) [132]. The authors suggest that the high toxicity associated with *Picea* CDs could be due to their physicochemical characteristics, such as smaller size (4.2 nm), which seems to be the main factor of toxicity of nanomaterials [133] (Table 3).

**Table 3.** Ecotoxicity of carbon dots.

| Carbon Dots | Ecotoxicological Effects | Reference |
|---|---|---|
| CDs made by electrochemical etching | *Chlorella vulgaris* biomass and growth rate increased by 17 and 21%, respectively. | [131] |
| CDs made by electrochemical oxidation | $EC_{50}$ for the growth of *Chlorella vulgaris* after 4 days of cultivation, based on the growth rate, was 70 µg mL$^{-1}$. | [130] |
| Biochar-derived CDs from *Picea, Molinia caerulea* and *Elaeis guineensis* | $EC_{50}$ (15 min) of 23.5 mg/L, 419.2 mg/L, and 8586 mg/L for *Picea, Molinia caerulea,* and *Elaeis guineensis* CDs, respectively. | [132] |
| $NaNO_3/KNO_3/NaNO_{2-}$ and citric-acid-based CDs | 40 mg/L CDs treatment inhibited the growth of the primary root of *Arabidopsis* greatly. | [134] |
| Amphoteric nitrogen-doped carbon dots (N–CDs) | N–CDs significantly alleviated the toxicity caused by high Cd stress on *Arabidopsis thaliana* seedlings growth. N–CDs induced higher germination rate (maximum: 2.5-fold), higher biomass (maximum: 3.7-fold), and better root development (maximum: 1.4-fold). | [135] |
| Citric-acid- and urea-based CDs | CDs at 250 and 500 mg/L showed no toxicity to maize (*Zea mays* L.). However, 1000 and 2000 mg/L CDs significantly reduced the fresh weight of root by 57% and 68% and decreased shoot fresh weight by 38% and 72%, respectively. | [136] |

Phytotoxicity assays revealed that CDs inhibited root growth by disrupting auxin biosynthesis and transport in *Arabidopsis* [134]. The CDs which were made of $NaNO_3/KNO_3/NaNO_2$ and citric acid also significantly inhibited *Arabidopsis* growth by 55%, as well as cell division and cell elongation, in root tips at 40 mg/L [134]. Conversely, N–CDs were investigated as the promotor and protector of plants under cadmium stress [135]. Nitrogen-doped CDs

were shown to be effective nanomaterials in enhancing Cd bioaccumulation and mitigating plant growth retardation in *Arabidopsis thaliana* seedlings under different levels of Cd stress. Moreover, the potential application of N–CDs to enhance plant absorption and immobilisation of $Cd^{2+}$, which could prevent its leaching into the environment, increase plant uptake efficiency, and alleviate any damage caused by $Cd^{2+}$ on plants, has been demonstrated, enhancing the total capacity of phytoremediation [135].

Other authors have assessed the effects of water-soluble CDs in maize (Chen et al., 2016). The study demonstrated that CDs at concentrations of 250 and 500 mg/L showed no toxicity to maize. However, at higher concentrations, such as 1000 and 2000 mg/L, CDs significantly reduced the fresh weight of root by 57% and 68% and decreased the shoot fresh weight by 38% and 72%, respectively [136]. Several authors suggest that carbonaceous nanomaterials have little or no bioaccumulation capacity, so there would be no chronic toxicity [133]. Future work is required to further clarify these findings.

In remediation studies, evaluating the toxicity of the effluents treated by nanomaterials is as important as examining the nanomaterials toxicity solely. Nevertheless, very little has been reported on the final toxicity of effluents treated using carbon-based materials, especially CDs, and whether they can significantly reduce the toxicity of such effluents. This is an important feature since the toxicity evaluation of industrial effluents may predict the adverse effects on the environment and their fate [137].

To investigate the potential risks of the photocatalytic degradation products of indomethacin (IDM), which is a non-steroid anti-inflammatory drug, acute toxicity evaluations were conducted through the detection of the changes in the emissions of the luminescent bacteria *Aliivibrio fischeri* using a Microtox™ assay, prior to and following treatment by photocatalyst N-doped $CDs/g-C_3N_4$ composites at $1 \text{ g L}^{-1}$ [138]. The initial inhibition rate of $4.0 \text{ mg L}^{-1}$ IDM was found to be 56.7%. During the photocatalytic treatment, the inhibition rate of treated solutions decreased to 36.7%. Subsequently, the toxicity increased and attained an inhibition rate of 43.2% when the irradiation duration was 45 min. This increase in toxicity may be attributed to the generation of additional toxic products during the photocatalysis of IDM. However, with further irradiation, the toxicity sharply decreased due to mineralisation. Similar results were found by Zhang et al. [139], where photocatalytic activity of a $CDs/BiPO_4$ composite was tested for IDM and the toxicity to other trophic levels was also investigated, using *Daphnia magna* (48 h) and *Chlorella* (96 h) [139]. Under prolonged irradiation, the acute toxicity of the treated solutions significantly decreased due to mineralisation.

Iron and nitrogen containing CDs (Fe, N–CDs) can catalyse the Fenton-like degradation of MB while minimising the effluents toxicity [113]. We have previously demonstrated that the application of Fe, N–CDs as catalysts under mild conditions showed promising degradation results for treating wastewater containing MB, resulting not only in a decolourised effluent (almost 100% in only 60 min) but also significantly reducing the ecotoxicity. Therefore, these early-stage studies indicated that for safe water treatment, an appropriate CDs approach could be applied along with ecotoxicity assays to monitor the treatments efficiency in terms of reducing any ecotoxicities for a potential safe release in the environment.

### 2.8.2. Microbial Responses to Carbon Dots in Soil and Water

The response of environmental microbial community composition and abundance following exposure to CDs is underexplored. The aquatic and soil ecosystems are likely to be the ultimate recipients for nanomaterial deposition; therefore, there are many questions that need to be addressed before considering their release in the environment. Microorganisms are key players in water and soil functioning and health as they are responsible for the cycling of mineral nutrients and the balance of a healthy environment. Since CDs are small, with diameters less than 10 nm, they can enter almost all areas of microorganisms. Guan et al. [85] developed a N-doped-CDs-supported carbon paper anode to enhance electricity generation of microbial fuel cells for wastewater treatment. The 16S rDNA pyrosequencing

analysis showed that the quantity of *Pseudomonas* on the electrode increased by ~40% compared with the electrode without modification, suggesting that the presence of N-doped CDs can alter the bacterial profile of the medium [85].

In soil treatment, CDs showed that lower concentrations could have a stimulatory effect on the growth of soil microbes [140]. The study revealed that soil treated with various concentrations of CDs (0.0–1000 µg/g) incubated for 90 days showed denaturing gradient gel electrophoresis (DGGE) profiles for PCR-amplified 16S rDNA consistent in soil samples from all treatments, regardless of treatment with CDs. Only some dominant bands in the DGGE profiles displayed some changes. The authors suggested that the introduction of CDs changes soil structure via restraining the growth of some sensitive bacteria in soil, and the CDs have less impact on the structure, function, and diversity of the soil microbial community when compared to semiconductor quantum dots (QDs) [140].

### 2.9. Challenges and Future Work of Carbon Dots in the Remediation of Dye-Containing Water

Even though CDs have been proven to show excellent application prospects in catalysis, there are still many challenges associated with their use requiring further study. For example, surface charge and chemistry of the dots affect their catalytic performance. In this review, we identified studies in which positively charged molecules were successfully degraded upon catalytic activity of negatively charged dots in the presence of $H_2O_2$ [100,114]. Although there is some discussion around their mechanism of action in Fenton-like reactions, there are still significant gaps in knowledge to be addressed, especially the efficacy of CDs on other molecules with distinct charges. In addition, according to the literature, metal-ion-doped CDs seem promising in enhancing catalytic behaviour; however, there is a lack of studies elucidating their mechanism.

As with other nanomaterials/AOPs systems, significant hurdles must be overcome before smaller scales can be translated to dye–wastewater treatment practices. The studies reviewed in this paper comprise laboratory-based studies where the catalytic activity of the dots was analysed under optimised lab-controlled conditions; nonetheless, further work aiming at larger scales with wastewater and real environmental samples are necessary to assess the actual efficacy of the CDs. In addition, the technoeconomic feasibility of the considered process should be investigated in future research.

To date, ecotoxicological studies have revealed conflicting results showing both positive and negative effects of CDs; therefore, more research is necessary, including focusing on the evaluation of the ecotoxicity of effluents treated by carbon dots, since insufficient investigations have addressed these important issues. Furthermore, studies on the influence of CDs on bacterial abundance and community structure in water and soil microbiota by next-generation sequencing have not been fully explored. Changes in microbial diversity may aid in the understanding of the effects of the eventual disposal of this emergent nanomaterial in the environment.

### 2.10. Conclusions and Future Perspective

In this critical review, we investigated whether CDs could act as efficient and eco-friendly catalysts for the remediation of dye-containing/contaminated water. As part of this, the key properties and characterisation techniques used in the assessment of CDs have been examined. The results of this review revealed that carbon dots can act as catalysts for the activation of $H_2O_2$ in Fenton-like reactions in the degradation of dyes in water under various conditions. Carbon dots can act as catalysts because of their inherent ability to accept and donate electrons during oxidation reactions, which could be improved through surface functionalisation with doping of specific chemical groups to create more active sites and electron density, as discussed in this review. Moreover, the performance of CDs can be improved depending on several environmental factors, such as temperature, pH, catalyst, and $H_2O_2$ concentrations, as identified in this review. However, ecotoxicological studies revealed conflicting results showing both beneficial and detrimental effects of CDs; therefore, more ecotoxicological research is necessary, focusing on the ecotoxicity

of the effluents treated by CDs. Some challenges in using CDs as catalysts have been identified, such as the need for optimised factors for a suitable catalytic performance, the need for investigations at a large scale, the potential toxicity of CDs when discarded in the environment, and more detailed information on the microbial dynamics affected by the CDs.

**Author Contributions:** Conceptualization, S.A.B., I.C. and A.S.B.; methodology, S.A.B., I.C. and A.S.B.; investigation, S.A.B.; writing—original draft preparation, S.A.B.; writing—review and editing, S.A.B., I.C. and A.S.B.; supervision, I.C. and A.S.B. All authors have read and agreed to the published version of the manuscript.

**Funding:** This research received no external funding.

**Acknowledgments:** We would like to thank the Brazilian National Research Council (CNPq) for the PhD Scholarship granted to Sabrina A. Beker (process number #200697/2017-0/GDE).

**Conflicts of Interest:** The authors declare no conflict of interest.

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
