# Peer review of "A Review on the Catalytic Remediation of Dyes by Tailored Carbon Dots"

_water, doi:10.3390/w14091456_

Round 1
Reviewer 1 Report
This manuscript deserves publication, however, major revisions are needed.
There is increasing interest in technologies for water and wastewater treatment since it has some inherent advantages over conventional chemical treatments. This is true especially for applications in developing nations since they are under different pressures compared with industrialized Western nations.
This manuscript summarizes literature information on CD for the removal of specific classes of contaminants such as dyes. It also provides information on the fundamentals of CD, and state-of-art as advanced oxidation process and catalysts.Authoritative review should critically consider previous publications and provide deeper understanding of underlying phenomena. Through the critical review, it should identify gaps in our current understanding thereby providing directions for future research. Unfortunately, this manuscript largely fails in this aspect, mainly comparing the results obtained when CD are used to degrade dyes. They do not even discuss whether earlier results are correct or not! Efficiencies of degradation are not discussed, mechanisms and limitations for each one of the works published and summarized. Also, "efficiencies values" in Table 2 are not clear. What is the parameter that has been considered here, total organic carbon, chemical oxygen demand or pollutant concentration removal?
A discussion about the works published using/testing CD in real wastewaters or water matrices should be given or indicate that it is a new direction or new challenge to be accomplished by the research.
Techno-economic analysis could be added, and commented.
Author Response
Dear Reviewer,
Please see the attachment.
Best regards,
The authors

Reviewer 2 Report
The authors focused on the literature survey of the dyes remediation using tailored carbon dots. The structure and the outline of the review article are very well. However, the authors need to add and cite further literature to support their review work.
Introduction section:
Please, take care of the spaces in the cited references. For instance,
(Liu et al., 2018;
(Xu et al., 2004
You need to update the water pollution paragraph with the following reference:
https://doi.org/10.1186/s42834-022-00131-0
Table 1 is not indicative of the advantages and disadvantages of each treatment technique. Please, add further points for each treatment. Please, use the following references:
https://doi.org/10.3390/w13050656
https://doi.org/10.1007/978-3-319-40148-5_7
Fig. 3 What is the source of TEM images of CDs?
Please, highlight the applicability of CDs in real wastewater.
What is the techno-economic feasibility of the synthesis of CDs?
I recommend the authors update their literature with the following references:
https://doi.org/10.1016/j.chemosphere.2021.132091
https://doi.org/10.1016/j.powtec.2020.02.073
https://doi.org/10.1016/j.jenvman.2020.110911
https://doi.org/10.1007/978-981-16-4823-6_2
https://doi.org/10.1021/acssuschemeng.8b00147
Author Response

(The authors gave the same response as above.)

Reviewer 3 Report
In this paper, the authors tried to cover the recent research advances in water remediation with the aid of carbon dots. The article is well written and well arranged. I suggest a minor revision before publication in Water.
- There are several reports on Carbon dots-based photocatalytic composites. However, this information is totally missing in the manuscript. The authors can add a paragraph covering C-dots with various semiconductor (metal oxides) materials.
- The conclusion section should be Conclusions and future perspective.
Author Response

(The authors gave the same response as above.)

Round 2
Reviewer 2 Report
The authors have addressed the most comments. I recommend accepting it.